# Humanin and Its Pathophysiological Roles in Aging: A Systematic Review

**DOI:** 10.3390/biology12040558

**Published:** 2023-04-06

**Authors:** Donatella Coradduzza, Antonella Congiargiu, Zhichao Chen, Sara Cruciani, Angelo Zinellu, Ciriaco Carru, Serenella Medici

**Affiliations:** 1Department of Biomedical Sciences, University of Sassari, 07100 Sassari, Italy; 2Control Quality Unit, Azienda-Ospedaliera Universitaria (AOU), 07100 Sassari, Italy; 3Department of Chemical, Physical, Mathematical and Natural Sciences, University of Sassari, 07100 Sassari, Italy

**Keywords:** senescence, humanin, aging and diseases

## Abstract

**Simple Summary:**

Humanin is a small mitochondrial-derived peptide still under study, with a number of potential therapeutic applications for various age-related diseases. It plays an important role in the body against several pathophysiological conditions; for instance, humanin has been shown to exert neuroprotective effects, which may make it a promising therapeutic candidate for Alzheimer’s and Parkinson’s disease. Moreover, it can help reduce inflammation and oxidative stress, thus interfering with the development of cardiovascular conditions and autoimmune diseases. Humanin also helps maintain the proper functioning of mitochondria, the dysfunction of which can contribute to a number of health problems, including neurodegenerative diseases and metabolic disorders, such as diabetes and obesity. It has been shown to improve glucose metabolism and insulin sensitivity in animal models. Treatment with humanin on mice is able to increase both their lifespan and health span. The mechanisms behind the protective effects of humanin in different diseases are slowly being unveiled and appear to be connected with autophagy and cytoprotective activity.

**Abstract:**

Background: Senescence is a cellular aging process in all multicellular organisms. It is characterized by a decline in cellular functions and proliferation, resulting in increased cellular damage and death. These conditions play an essential role in aging and significantly contribute to the development of age-related complications. Humanin is a mitochondrial-derived peptide (MDP), encoded by mitochondrial DNA, playing a cytoprotective role to preserve mitochondrial function and cell viability under stressful and senescence conditions. For these reasons, humanin can be exploited in strategies aiming to counteract several processes involved in aging, including cardiovascular disease, neurodegeneration, and cancer. Relevance of these conditions to aging and disease: Senescence appears to be involved in the decay in organ and tissue function, it has also been related to the development of age-related diseases, such as cardiovascular conditions, cancer, and diabetes. In particular, senescent cells produce inflammatory cytokines and other pro-inflammatory molecules that can participate to the development of such diseases. Humanin, on the other hand, seems to contrast the development of such conditions, and it is also known to play a role in these diseases by promoting the death of damaged or malfunctioning cells and contributing to the inflammation often associated with them. Both senescence and humanin-related mechanisms are complex processes that have not been fully clarified yet. Further research is needed to thoroughly understand the role of such processes in aging and disease and identify potential interventions to target them in order to prevent or treat age-related conditions. Objectives: This systematic review aims to assess the potential mechanisms underlying the link connecting senescence, humanin, aging, and disease.

## 1. Introduction

By 2050, the population aged 85 and over will triple [1]. Understanding demographic trends is crucial for national development planning, for the implementation of the 2030 Agenda, and for sustainable development in which people are at the center [2]. Recent demographic trends are harbingers of future challenges for health systems and are of considerable importance for public health [3]. Aging-related diseases (ACVD) increase as the population ages, placing a heavy economic burden on society [1,2].

Aging is a complex biological pathway due to the synergistic effects generated by multiple processes that disrupt the structures and functions of an organism on different scales, going from the molecular, to the cellular, to the organ levels. The causes of aging include, but are not limited to, oxidative stress, glycation, telomere shortening, mutations, side reactions, epigenetic changes, protein aggregation, and many others [4]. Oxidative stress is just a part (but an important piece) of this pathway, and oxidative phosphorylation in the mitochondria is the main source of reactive oxygen species (ROS).

Increased levels of ROS disrupt the dynamic balance between the oxidative and antioxidant systems [5]. Many strategies, such as the study of ferroptosis [6], have evaluated suppression of ROS generation as a potential strategy against senescence. Among the molecules counteracting oxidative stress, polyamines, whose regulation is over-regulated by the ODC gene transcription, have also emerged and play multiple roles [7,8,9,10].

Overall, aging is a metabolic process characterized by loss of cellular homeostasis, increased susceptibility to chronic diseases, and altered resilience. Moreover, it is affected by diet (e.g., dietary restriction), genetics (e.g., insulin/insulin-like signaling), and pharmacological interventions (e.g., rapamycin, metformin) [11], resulting in metabolic, immune, and inflammatory remodeling, during which mitochondrial metabolism plays an important role. Until now, mitochondrial–nuclear communication has always been considered the key to healthy aging [12] and is mediated by core-encoded proteins, transient molecules, and mitochondrial metabolites; now, mitochondrial-derived peptides (MDP) are also emerging as key players through interactions with stress-sensitive transcription factors [13]. Among them, humanin in particular (*vide infra*), or better its lack, contributes to specific aging processes, including cellular senescence, chronic inflammation, and cognitive decline.

Mitochondria are the powerhouse and the beating heart of the cell, and mitochondrial dysfunction is implicated in many age-related diseases, the causal or simply related role of which is still unclear. Thus, in the last decade, mitochondrial-derived peptides and micro peptides have attracted the interest of many researchers primarily because they regulate mitohormesis, which is, in turn, related to lifespan [14]. MDPs cause several cellular effects such as the activation of nuclear genes and signaling pathways [15]. Naturally occurring MDPs are age-dependent regulators of apoptosis, insulin sensitivity, and inflammatory markers [14,16]. Humanin (HN) is a short peptide, as shown in Figure 1, consisting of 21 or 24 amino acids (depending on mitochondrial or cytoplasmic translation, respectively), and it was discovered in 2001 by Nishimoto Lab [17,18] during a screening of neurotrophic factors that inhibited apoptosis induced by a pathological amyloid beta mutant in Alzheimer’s disease [19]. It is one of the nine (and the first to be discovered) MDPs, each of which has been shown to have various cytoprotective or metaboloprotective properties, encoded by short ORF (open reading frame) sequences located within the 16S ribosomal RNA gene MTRNR2. The humanin gene is presumed to be an ancient mitochondrial signal used to communicate with the rest of the organism; in fact, it is highly conserved even in distant species such as the nematodes [20]. An interesting aspect of humanin is that it possesses biological activity and does not have a signal peptide for secretion, but may act as a signal peptide itself [21].

To date, it is well established that humanin is expressed in different tissues of various organs, including the heart, kidney, liver, testes, skeletal muscle, and brain [22]. It acts at the cellular level, but can be found secreted into plasma, cerebrospinal fluid and seminal fluid, although it is still unclear which tissue(s) contribute to the circulating pool of this peptide and which is the secretory mechanism of its release. Entry into cells is even less known. Thus, the role of humanin has not yet been fully clarified; what is clear is that its involvement in the protection against many other age-related diseases such as stroke and atherosclerosis has expanded with various beneficial consequences now becoming evident [23,24]. HN has multiple interactors, not all of which are probably known. Some experiments have revealed how this peptide could interact, by inhibiting it, with the pro-apoptotic protein BAX, a member of the Bcl-2 family of genes which regulate programmed cell death, involved in apoptosis [20,25,26]. At the same time, P. Cohen found that HN interacts with insulin-like growth factor (IGF)-binding protein-3 (IGFBP-3) and possesses both growth-inhibiting and enhancing effects on cells [27]. It acts as a ligand for two different membrane receptors: FPRL1 and G-protein-coupled, for intracellular calcium mobilization and activation of the ERK1/2 pathway, implicated in cell survival responsible for cytoprotective effects. In addition, it exerts cytoprotective and antioxidant effects from interaction with the trimeric extracellular receptor CNTFR/gp130/WSX-1, which leads to phosphorylation of JAK2 and STAT3 and transcription of genes conferring cryoprotection. Humanin is also able to stimulate chaperone-mediated autophagy (CMA) which, by directing oxidized proteins to the lysosomes, helps maintain quality control and confers cytoprotection [28]. Circulating levels of humanin, according to conflicting studies, decrease with age in mice and humans [29,30]. This could be due to the age-dependent accumulation of mitochondrial DNA (mtDNA) damage, including deletions and point mutations [31], as well as a decrease in mitochondria number and mtDNA copy number in some organs [32]. Kim et al. hypothesize that humanin but also other MDPs may act to attenuate the senescence phenotype or even function as senolytic agents themselves in cell culture experiments [33]. Not only wild-type HN, but also some of its synthetic derivatives have demonstrated to be highly active, as in the case of HN-S14G (HNG), where the serine at amino acid residue 14 was substituted with a glycine. HNG was more effective than HN on the reduction of intracellular ROS, as well as in the preservation of mitochondrial membrane potential and structure with cardioprotective activity in vivo [34,35], for instance. This, and many other aspects connecting humanin and senescence will be examined hereafter, as an attempt to unveil the mechanisms involved.

**Figure 1 biology-12-00558-f001:**
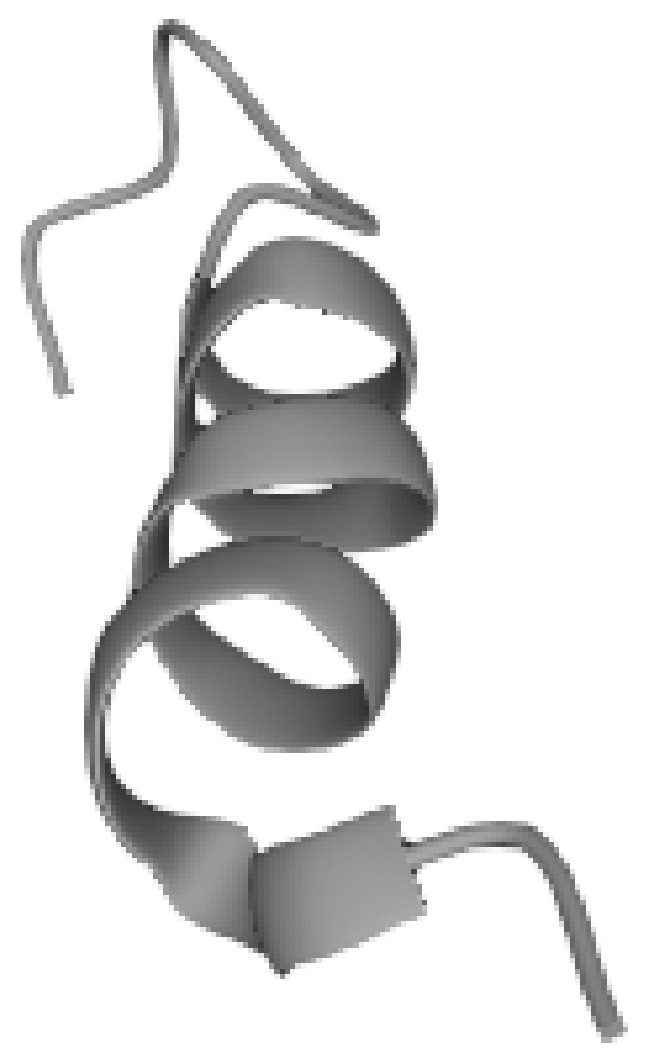
Humanin molecule, whose sequence is: Met-Ala-Pro-Arg-Gly-Phe-Ser-Cys-Leu-Leu-Leu-Leu-Thr-Ser-Glu-Ile-Asp-Leu-Pro-Val-Lys-Arg-Arg-Ala [36].

## 2. Materials and Methods

This review was undertaken under the guidance of Preferred Reporting Items for Systematic Review and Meta-analysis, Figure 2. Regarding the Institutional Review Board Statement and Informed Consent Statement, they are not applicable.

### 2.1. Search Strategy

A comprehensive search of multiple databases was conducted to identify relevant studies. The databases searched included Google Scholar, PubMed, and Scopus. The search included published and unpublished studies and was conducted in December 2022. The search terms used included ‘senescence’, ‘humanin’, ‘aging’, and ‘disease’. The search was limited by the English language.

### 2.2. Inclusion Criteria

This review considered studies investigating senescence and/or humanin in any tissue or organ as they satisfied the eligibility criteria. The other considerations in the inclusion criteria were studies that focused on human or animal subjects or cells.

### 2.3. Exclusion Criteria

This review considered exclusion studies that failed to investigate senescence and/or humanin. The review excluded editorial comments and conference abstracts.

### 2.4. Data Extraction

Two independent reviewers using a standardized form participated in the data extraction process. The following information was extracted from each study: study design, participant characteristics such as gender, age, interventions, disease status, sample size, outcomes, and results. Any discrepancies between the reviewers were resolved through discussion and consensus. The study characteristics were then tabulated.

### 2.5. Quality Assessment

The quality of the included studies was assessed using the ROBIS tool for assessing the risk of bias. Two independent reviewers performed the quality assessment, and any discrepancies were resolved through discussion and consensus. The overall quality of each study was rated as ‘low risk’, ‘high risk’, or ‘unclear risk’ based on the ratings of the individual domains.

### 2.6. Analysis

Senescence and humanin are involved in the aging process and the development of age-related diseases. Senescence occurs when there is a decay in cell function and proliferation, with an increase in cell damage and death. On the other hand, humanin is defined as a cytoprotective peptide of 24 amino acids encoded by short open reading frames within the mitochondrial genome.

Both senescence and humanin have been linked to several diseases, including neurodegeneration, cardiovascular disease, and cancer. However, the exact mechanisms underlying the link between these processes, aging, and disease are not yet fully understood. This review aims to assess the current evidence on the potential mechanisms underlying the link between senescence, humanin, aging, and disease.

## 3. Results and Discussion

The relatively recent discovery of humanin, together with the necessity of clarifying its role and functions, are responsible for the so far limited number of studies where this mitochondrial peptide has been directly connected to senescence, while others deal with different dysfunction that can lead to or are related with aging processes. Nevertheless, the literature reports various interesting and enlightening papers on the topic, as follows.

### 3.1. Brain and Neurodegeneration

A decay in mitochondrial activity and quality has been related to impairment of mitochondrial membrane potential, which is the central bioenergetic parameter controlling ATP synthesis, respiratory rate, and generation of ROS correlated with the development of different age-related diseases such as neurodegeneration. Kim [37] and other researchers focused on HN playing a crucial protective role against cognitive decline and associated neurological conditions such as Alzheimer’s disease. The molecular pathway implicated in cognitive function is complex and interferes with the release of cytochrome, which, in turn, interacts with the pro-apoptotic proteins Bim, Bid, and Bax. Furthermore, humanin localizes onto the membrane surface of lysosomes, therefore boosting the activation of chaperone-mediated autophagy, leading to the degradation of misfolded proteins, suppressing fibril formation and Aβ-induced toxicity through the activation of pro-survival signaling cascades associated with kinases, including AKT1 kinase, tyrosine kinases associated with STAT3 and caspase-3 activation, and extracellular signal-regulated kinases 1 and 2 (ERK1/2). A considerable protective action in cardiovascular conditions has also emerged, and treatment with HNG caused the reversal of cardiomyocyte apoptosis in aged mice, together with the decrease in fibroblast proliferation and collagen deposits. These authors also reported that such effects are mainly due to HN capability to reduce mitochondrial dysfunction and discussed the correlation between humanin and aging in two animal models (the worm *C. elegans* and mice). Humanin decreased functional cognitive decline and overall inflammation induced by IBA-1, IL-6, and IL-10. Moreover, HN treatment induced positive results on metabolic aging, as it decreased midlife adiposity by increasing lean body mass and reducing visceral fat [38]. Finally, HN was able to produce a small, but significant, expansion of the lifespan in transgenic worms. Such an effect was dependent on daf-16/FOXO, indicating that the insulin/IGF signaling pathway is critical for this mechanism [39].

Yen et al. in 2018 demonstrated the neuroprotective effect of humanin, both in in vitro and in vivo experiments [38]. The protective effect of HN and its analog HNG against amyloid-beta (Aβ) mitochondrial toxicity was tested in vitro. SH-SY5Y cells, a human neuroblastoma cell line, with soluble Aβ, were treated with and without HNG. The authors showed that administration of humanin prevented some of the normal age-related behavioral and cognitive deficits in common laboratory mice. It was also observed that plasma humanin levels were reduced in African Americans, linking the mitochondria and mitochondrial peptides to health disparities [38]. Later, the same authors [40] analyzed the levels of humanin in different living forms, showing, for instance, that humanin treatment conferred protection against lethal heat-shock doses in yeasts. They also generated a transgenic *C. elegans* worm, overexpressing humanin, and noted that such overexpression correlated with an increase in lifespan dependent on the daf-16/FOXO gene. This resulted in reduced body size, body fat, and reproductive output in the worms. A similar effect was found in humanin transgenic mice, which showed a 16% increase on circulating humanin and were protected against toxic insults. In another experiment, middle-aged mice underwent exogenous administration of HNG to twice a week. This treatment reduced inflammatory markers and improved metabolic health parameters. In addition, the authors analyzed circulating levels of humanin in a cohort of human centenarians, but due to the lack of a control group, they rather examined their children, who had indeed higher levels of humanin compared to control subjects. Analyzing humanin levels, the researchers found that mtDNA copy numbers decrease in human conditions such as Alzheimer’s disease and MELAS (mitochondrial encephalopathy, lactic acidosis, and stroke-like episodes). Together, these studies showed that humanin is linked to better health and a longer lifespan [40].

Lee et al. also made a digression on the regulation by mitochondria of metabolic homeostasis at the cellular and organismal level via humanin, suggesting the possible existence of additional sORFs in mtDNA [41]. Humanin and its homologs including HNG, which is 1000 times more potent, block memory impairment associated with the neurotoxic scopolamine, 1,4–6 Aβ (Amyloid-β), and 3-quinuclidinyl benzilate [33]. In another study by Caso et al., the interaction of humanin with formylmethionine-leucyl-phenylalanine (f-MLP) FPRL1, FPRL2 protects neurons from Aβ and is STAT-dependent [42]. Furthermore, in mouse models, it reduced plaque accumulation, improved cognitive dysfunction by acting on the prefrontal cortex in patients with Alzheimer’s disease and has a cytoprotective action in stroke [43], and acts positively in a model of familial Amyotrophic Lateral Sclerosis (ALS) combined with neurotrophic factor [44]. Its action is not the same as in other neurodegenerative diseases such as prion disease and Huntington’s disease [45]. Additionally, in the mouse model, humanin has a cytoprotective action in diabetes [46], atherosclerosis [47,48], and in myocardial infarction and reperfusion injury [49], atherosclerosis, by maintaining endothelial cell function, and in hypercholesterolemic subjects, ApoE blocks atherosclerotic plaque progression [33].

Zárate et al. studied the neuroprotective effect of humanin [50]. Ovarian hormones, particularly estrogen, are regulators of mitochondrial function, which is crucial in organs and tissues with a high energy demand such as the central nervous system. In a rat model of surgical menopause, the effect of ovarian hormone deprivation on humanin expression and localization in the hippocampus was studied. Humanin expression was lower in the hippocampus of ovariectomized rats, and its immunoreactivity was colocalized with astroglial markers. Chronic ovariectomy promoted structural changes in hippocampal astrocytes, which are directly proportional to reduced HN expression in this brain area. Ovarian hormones increased the intracellular content and release of humanin by astrocytes in culture. Humanin prevented glutamate-induced dendritic atrophy and reduced puncta number and total puncta area for the pre-synaptic marker synaptophysin in cultured hippocampal neurons.

As other authors, Table 1, have already pointed out, the effect of HN on Alzheimer’s disease has been studied both in vitro and in vivo [51]. HN suppresses neuronal cell death induced by Aβ and three different types of familial AD (FAD) genes including presenilin (PS)1 and PS2, and mutant APP. Moreover, HN inhibits neurotoxicity by AD-relevant insults induced by other FAD genes including A617G-APP, L648P-APP, A246E-PS1, L286V-PS1, C410Y-PS1, and H163R-PS1, and other Aβ peptides (Aβ 1–42 and Aβ 25–35) [52]. The potent HN analog HNG not only inhibits the formation of the Aβ 1–42 fibrils but also causes disaggregation of the preformed fibrils [53]. HNG inhibits Aβ 1–42 fibril formation, disaggregates preformed fibrils, and protects against Aβ-induced cytotoxicity in vitro [54]. Administration of HNG with a 3-month IP injection treatment significantly improves spatial learning and memory deficits, decreases Aβ plaque accumulation and insoluble Aβ concentrations, and reduces neuro-inflammatory responses in middle-aged APPswe/PS1dE9 mice, a double transgenic mouse model of AD overexpressing APP, and mutant human PS1 in neurons [55]. The use of hyperinsulinemic–euglycemic clamp and hyperglycemic clamp techniques helped clarify the role of HN in glucose homeostasis. The crucial action of HN appeared to be mediated by the activation of hypothalamic STAT-3 signaling pathway. Continuous IV infusions of HNGF6A (another HN analog that is potent, stable, and non-IGFBP-3 binding) during the hyperinsulinemic–euglycemic clamp, suppressed hepatic glucose production and increased GIR, peripheral glucose uptake. Finally, HNGF6A significantly decreased the blood glucose in Zucker diabetic fatty rats when administered as a single IV injection.

### 3.2. Heart and Cardiovascular Diseases

Evidence is emerging that serum levels of HN are negatively correlated with age and related to age-connected cardiovascular disease, Table 2, [29,40,56,57].

Experimentally, it was found that HN, by promoting the expression of antioxidant defense system proteins, reduced oxidative stress induced by H_2_O_2_ and reduced ROS production by restoring chaperone-mediated autophagy in myocardial cells and myocardial mitochondria [34,58,59]. HN upregulates antioxidant enzyme expression, preserving cardiac function after myocardial infarction in an ischemia-reperfusion injury model by reducing myocardial cell death and myocardial infarct area [18,19,20]. Furthermore, HN protects, in vitro, endothelial cells from oxidation produced by ROS derived from abnormal glycolipid metabolism [60,61,62].

Humanin acts in age-related ventricular stiffness and heart failure by reducing the process associated with cardiac fibrosis. Moreover, in adult mouse models, it reversed the ratio of cardiomyocytes to fibroblasts to levels similar to those measured in young 5-month-old mice by inhibiting cardiomyocyte apoptosis and fibroblast proliferation [63]. Recently, it was discovered that humanin has six similar peptides encoded by the 16S RNA gene, which also have an action on autophagy chaperones (CMAs) to maintain protein control and direct oxidized proteins to the lysosome [64]. Moreover, circulating levels of humanin were significantly higher in long-lived Ames dwarf mice, but lower in short-lived growth hormone (GH) transgenic mice linking it to longevity [65]. Furthermore, there is a negative correlation with GH/IGF-1, and treatment of mice and humans with GH or IGF-1 reduces their circulating levels. Humanin would be expected to increase with age, correlating with an increase in cellular stress associated with age-related pathology; in fact, Conte et al. directly contradict the data reported so far by examining a human cohort of 693 subjects aged 21 to 113 years [30].

Gong Z et al. took up the basic concepts of MDPs, and in particular of humanin [49]. It emerges from the data that HN influences various metabolic processes including autophagy, ER stress, cell metabolism, inflammation, and oxidative stress, as well as playing critical roles in aging, various metabolic disorders, and cardiovascular and autoimmune diseases. They analyzed and collected data on its role in cardio-protection after myocardial ischemia-reperfusion, associated with a significant increase in ROS-related damage to the mitochondrial respiratory chain machinery causing further ROS production and a vicious cycle of bioenergetic decline that ultimately leads to cell death. After an ischemic damage, myocardium undergoes remodeling that includes cardiac hypertrophy, extracellular matrix deposition and excessive myofibroblast proliferation, and to meet the blood supply demand. A study conducted by RH Muzumdar et al., showed that intracardiac treatment with a single dose of HNG, either at the time of ischemia or reperfusion, improves cardiac function and decreases infarction size in a mouse model [66,67]. Exogenously administered HNG increased cardiomyocyte survival and inhibited apoptosis. Thummasorn et al. found that HNG treatment decreased cardiac arrhythmias and infarct size and improved the function of the left ventricle during MI-R [68,69,70]. The authors showed that HNG treatment enhanced cardiac mitochondrial function and decreased the mitochondrial ROS level in cardiac cells. Thomas E. Sharp III et al. translated the cardio protection seen in mice to a porcine model of MI-R, which is a model very similar to humans in terms of myocardial structure, cardiac metabolism, and the coronary artery anatomy [71]. Results showed that a single dose of HNG, administered by intravenous injection at the time of reperfusion, inhibits cardiomyocyte apoptosis and decreases cardiac infarction size in a porcine model of MI-R with 60 min of ischemia. The protective effects were nullified when the animals were subjected to 75 min of ischemia, indicating that the ability of HN to limit MI-R-induced infarction depends on the duration of myocardial ischemia, and consequently, the extent of injury. It was also observed that patients with coronary heart disease show lower circulating HN levels compared to healthy controls. HN levels are positively associated with homoarginine, and negatively correlated with circulating levels of lactic acid, the latter indicating a relationship with mitochondrial oxidative metabolism. Subjects with angina and MI had lower circulating HN levels compared to controls, and the HN levels could be exploited as a prognostic biomarker for major cardiac events in patients with angina.

**Table 2 biology-12-00558-t002:** Role of Humanin in cardiovascular disease, N/A is not applicable.

Article	Study Design	Population	Outcome Measures
Yen, K. et al. (2020) [40]	In vivo study	*C. elegans*, Mouse, Human	Circulating levels of humanin and their relation to diseases of aging and lifespan
Cai, H. et al. (2021) [72]	Review	N/A	Protective effect of humanin against oxidative stress
Conte, M. et al. (2019) [30]	In vivo study	Human	Aging; longevity. plasma levels of different mitokines
Gong, Z. et al. (2022) [49]	Review	Mouse, Porcine	Protective effect of humanin in myocardial ischemia-reperfusion.
Muzumdar, R.H. et al. (2010) [67]	In vivo study	Mouse	Intracardiac administration of a single dose of HNG at the time of ischemia or reperfusion reduces infarction size and improves cardiac function. Humanin attenuated protein levels of Bax in the heart following MI.
Thummasorn, S. et al. (2016) [68]	In vivo study	Mouse	HNG treatment reduced cardiac infarct size, improved the function of the left ventricle, and decreased cardiac arrhythmias during MI-R. HNG treatment improved cardiac mitochondrial function and decreased the mitochondrial ROS level in cardiac cells
Sharp, T.E., III et al. (2020) [71]	In vivo study	Porcine	A single dose of HNG, administered at the time of reperfusion, diminishes cardiac infarction size and inhibits cardiomyocyte apoptosis

Several studies demonstrated that humanin is able to inhibit MI-R-induced apoptosis in both porcine and murine models. The cytoprotective role of HN is related to its anti-apoptotic effect. Muzumdar et al. reported that HN decreased the levels of B-cell lymphoma-2 (Bcl2)-associated X protein (Bax) in the heart after MI [67] in a mouse model of MI-R. This is due to the fact that humanin directly binds to and inactivates the apoptosis-inducing protein Bax [20] and extra-long isoform of Bim (BimEL). Binding of HN to insulin-like growth factor-binding protein 3 (IGFBP3) has also been reported, together with importin-β1 interaction and translocation and prevention of IGFBP3, resulting in a reduction of apoptosis. Furthermore, HN can also alter apoptosis through regulation of signaling cascades. HN binds to multiple cell surface receptors, such as ciliary neurotrophic factor receptor alpha (CNTFR), formylpeptide receptor-like-1 molecule (FPRL-1), WSX-1, and gp130. It also activates signal transducer and activator of transcription (STAT) 3 protecting neuronal cells from Aβ-induced apoptosis. HN mitigates MI-R-induced oxidative stress in the cardiomyocytes by promoting the removal of oxidative stress-damaged proteins and the activation of antioxidant enzymes by switching on chaperone-mediated autophagy CMA. Finally, HN capability to promote glycolysis, glucose utilization, and ATP production in energy-scarce situations allow cardiomyocyte survival while limiting ROS production. All these mechanisms are able to stimulate cardiomyocyte survival and decrease MI-R-induced apoptosis.

### 3.3. Immune System and Inflammation

A characteristic feature of aging appears to be the presence of a large number of dysfunctional mitochondria, which may be the trigger of inflammatory stimuli and concur to immune impairment. Some authors, Table 3, speculate that mitochondrial dysfunction may be beneficial to the cell because it may elicit an effective stress response [73] leading to the activation of compensatory mechanisms that increase lifespan [74] through a process called mitohormesis. Mitohormesis is an adaptive mechanism whereby moderate stress stimulates the activation and increased production of antioxidant defense systems. Mild mitochondrial stress would increase the resistance and survival of the entire organism, not only limited to the affected tissue or cell, but also spread to distal areas [75], due to increased ROS formation, which, in turn, activates a retrograde stress-resistance response culminating in lifespan extension. These mitochondrial perturbations are spread throughout the body via soluble mediators, among these are catalogued peptides encoded by mtDNA that include humanin [76,77]. These molecules are called mitokines, secreted in response to a mitochondrial stress reaction, and they increase with age and participate in chronic inflammation termed inflammaging, supporting the idea of mitohormesis introduced by Ristow and Zarse [78]. The age-related increase in mitokines could therefore be considered as an attempt at mitohormesis aimed at improving stress resistance [30,79]; nonetheless, when the production of otherwise beneficial cytokines goes from acute to chronic, these become either ineffective or even toxic [30]. HN exerts a cytoprotective action against oxidative stress, apoptosis, and inflammatory response [80], while in immune responses, as an anti-inflammatory mediator, HN decreases macrophage infiltration and inflammation, as well as apoptosis, by interacting with the gp130 subunit of the IL-6 receptor, leading to a reduced in vitro production of pro-inflammatory cytokines, such as IL-1β, IL-6, and TNF-α [81,82,83,84]. HN has anti-apoptotic effects because it reduces apoptosis-related pro-inflammatory DAMP levels. However, the action of HN is linked to its interaction with IGFBP-3, a protein particularly expressed in the AD brain, with a role in cell death [27,85]. Furthermore, HN interacts with the apoptosis-inducing protein Bax, creating an HN-Bax complex, which, in the cytoplasm, prevents translocation into the mitochondria, protecting against apoptosis [86,87]. This mechanism is rather important, as the activation of Bax/Bak complexes causes the release of pro-apoptotic molecules, but also mtDNA and dsRNA that can, in turn, activate a pro-inflammatory response via inflammasomes [87]. HN has not only cytoprotective, but also anti-inflammatory roles in markedly inflammatory diseases such as in Alzheimer’s disease [88] but also in type 2 diabetes (T2D), cardiovascular disease (CVD), and atherosclerosis [51,82,87,89,90,91]. Available data, in a recent paper by Salemi et al., point to an anti-inflammatory role of cytokines and, in particular, humanin, which shows it to be significantly upregulated in fibroblasts from people with Down’s syndrome compared to their non-trisomic siblings [92] as well as in vivo in people with accelerated aging such as people with Down’s syndrome, compared to their siblings of similar age [93]. Finally, the authors summarize the proposals that HN, and other cytokines, can be used experimentally as an anti-inflammatory therapeutic tool and target, both in humans and in animal models [87].

Nashine et al. conducted a study aimed at demonstrating that treatment with HNG can reduce plasma levels of inflammation-associated marker protein which play a crucial role in the pathogenesis of AMD (age-related macular degeneration) [94]. HN levels were measured in the plasma of AMD patients and normal subjects using ELISA assay. A reduced level of humanin and an elevated level of inflammatory proteins were observed in the plasma of patients with AMD compared to control plasma samples. HNG was added to normal (control) and AMD RPE transmitochondrial cybrid cells, which had identical nuclei from mitochondria-deficient ARPE-19 cells but differed in mtDNA content derived from clinically characterized AMD patients and normal (control) subjects. Cell lysates were extracted from untreated and HNG-treated AMD and normal cybrids, and the levels of inflammatory proteins were measured with Luminex XMAP multiplex assay. In AMD RPE cybrid cells, treatment with HNG reduced CD62E/E-Selectin (by 64.62%), CD62P/ P-Selectin (by 60.99%), ICAM-1(by 30.82%), TNF-α (by 46.09%), MIP-1α (by 61.98%), IFN-γ (by 62.86%), IL-1β (by 68.31%), IL-13 (by 57.63%), and IL-17A (by 48.31%) protein levels. Therefore, HNG seems to act as a suppressor of the inflammatory proteins that may contribute to the development of AMD. Furthermore, it has been observed that in the cells that are healthy and have normal homeostasis, HNG does not exert any negative effect.

In another recent study by Conte et al., plasma levels of fibroblast growth factor 21 (FGF21), growth differentiation factor 15 (GDF15), humanin, and mitochondrial stress-related mitokines, were measured in patients with type 2 diabetes (T2D) or Alzheimer’s disease (AD) compared to healthy subjects, including centenarians’ offspring (OFF) [89]. Circulating FGF21 levels were higher in OFF and lower in AD but not in T2D patients, GDF15 levels were higher in T2D patients but not in AD ones, and HN levels were lower in both T2D and AD patients, particularly in the latter. As for T2D, all the three mitokines evidenced interesting associations with the presence of complications and they were related to worsening eGFR. No association was observed between HN and T2D. On the other hand, a negative correlation was found between HN with HbA1c and glycemia, suggesting that HN could be involved in the maintenance of insulin sensitivity.

Still, the same author, Conte et al., in a study carried out in 2019, measured plasma levels of different mitokines, in particular, fibroblast growth factor 21 (FGF21), growth differentiation factor 15 (GDF15), and humanin, in 693 subjects aged from 21 to 113 years and in 28 sibling pairs constituted by a person with Down’s syndrome (age range 11–43 years) and a non-trisomic sibling (age range 8–52 years), and analyzed the association of these mitokines with parameters of health status [30]. An increase in circulating levels of FGF21, HN, and GDF15 in old age was observed, with the highest levels found in centenarians. These molecules are associated with worsened parameters (such as triglycerides, insulin sensitivity, and handgrip strength), especially in 70-year-old subjects, and their levels are inversely correlated with survival in the oldest subjects. Persons with Down’s syndrome (characterized by accelerated aging) have higher levels of GDF15 and HN with respect to siblings [30].

Woodhead et all., also report changes in plasma humanin levels in long-lived subjects. The increase in plasma HN has been interpreted as a hormetic response, with either beneficial or detrimental effects [7], depending on the levels of response toward stress [95].

### 3.4. Diabetes and Obesity

Boutari et al. indicated that HN improves the survival of pancreatic beta cells, increases insulin sensitivity, and delays the development of diabetes [46]. HN is regulated by growth hormone (GH) and insulin-growth factor 1 (IGF-1). HN and IGF-1 levels decrease with age. It has also been proposed that HN levels are inhibited by GH via IGF-1. Circulating HN levels are reduced by treatment with GH or IGF-1 in both humans and mice. Concerning diabetes, HN confers protection against cytokine-induced apoptosis by binding pro-apoptotic Bax, inhibiting its mitochondrial localization, and diminishing Bax-mediated apoptosis activation, acting either through the FPRL-1 receptor or directly on Bax. As for neuroendocrine beta cells’ protection, it involves HN binding to a complex involving CNTFR/WSX-1/GP130 and activation of tyrosine kinases and STAT-3 phosphorylation. The interaction between HN and insulin-like growth factor-binding protein-3 (IGFBP-3) which prevents the activation of caspases, is also relevant in these mechanisms.

Moreover, both humanin and small humanin-like peptide 2 (SHLP2) have been reported to modulate metabolism and act as insulin sensitizers. A study conducted by Mehta et al. investigated how humanin and SHLP2 treatment is able to regulate the plasma metabolite profile in a diet-induced obesity (DIO) mouse model. In their experiments, plasma samples were obtained from DIO mice subjected to vehicle (water) treatment, or peptide treatment with either HNG or SHLP2. The latter significantly altered the levels of amino acid and lipid metabolites in plasma. In fact, the treatment with peptides led to the reduction of major intermediates in the glutathione cycle and sphingolipid pathways which play a key role in lipid signaling and a potential connection to insulin resistance and apoptosis [96].

Additionally, Muzumdar et al. conducted an in vivo study on 101 rats to investigate the role of central and peripheral HN on insulin action. It was demonstrated that continuous intra-cerebro-ventricular infusion of HN significantly improved overall insulin sensitivity [29,97]. Activation of hypothalamic STAT-3 signaling was associated with central effects of HN on insulin action; such effects were canceled by co-inhibition of hypothalamic STAT-3. Peripheral intravenous infusions of novel and potent HN derivatives reproduced the insulin-sensitizing effects of central HN. HN levels were also measured in rodents to examine whether there were age-related changes. The amount of detectable HN in the cortex, skeletal muscle, and hypothalamus diminished with age in rodents, while circulating levels of HN were decreased with age in mice and humans. This change suggests that HN could play a role in the pathogenesis of age-related diseases as reported in Table 4, [96].

### 3.5. Potential Mechanisms Involved in the Protective Effects of Humanin

#### 3.5.1. Autophagy

As previously seen, some authors explored several systems, including cell cultures, mice, *C. elegans*, and RNA-seq of the human brain, the mitochondrial functions and synthesis of humanin from mtDNA, which lacks repair mechanisms like those found in the nucleus [98]. This, combined with the proximity of mtDNA to the electron transport chain, gives rise to a high number of mtDNA mutations by interaction with reactive oxygen species. Mutations are correlated with aging, abnormal body composition, reduced fertility, and shortened lifespan [98,99]. Yet humanin and its homologs increased lifespan and mechanism are still unknown [45]. Currently, humanin is being closely examined for its relationship to metabolic decompensation-associated diseases such as diabetes and obesity, as well as attracting attention for its possible association with cancers due to the Warburg effect, which has already been correlated with mitochondrial dysfunction and mtDNA instability [100]. Humanin is now established to induce autophagy in transgenic worms and several cell types, and in skeletal muscle in mouse models, Table 5, [40]. Autophagy is activated to prevent the accumulation of dysfunctional damaged mitochondria because they could induce oxidative damage, muscle catabolism, or by excess autophagy, atrophy [101]. A hallmark of aging appears to be the presence of large numbers of dysfunctional mitochondria, which may be the source of inflammatory stimuli and contribute to immune impairment. Some authors speculate that mitochondrial dysfunction may be beneficial to the cell because it may elicit an effective stress response [72]. Through these, mitochondria regulate not only energy production but a cellular homeostasis that includes a range of processes such as immune/inflammatory responses, proteostasis, adaptive responses to stress, and apoptosis [96,97,98]. Muscle humanin expression is upregulated in humans, in response to exercise stress, mtDNA mutation-associated diseases, and healthy aging, suggesting a tissue-specific response aimed at restoring mitochondrial homeostasis [102] in response to stress from, for example, acute exercise [102]. Several skeletal muscle diseases are manifested by either an excess of leading to atrophy, or a reduced autophagy mechanism leading to muscle and mitochondrial degeneration [103].

The authors dwell on the elevated expression of humanin in muscles of patients with the mitochondrial mutations leading to MELAS (mitochondrial encephalomyopathy with lactic acidosis and stroke-like episodes) and CPEO (chronic progressive external ophthalmoplegia) [104,105]. Furthermore, in patients with chronic kidney disease, circulating humanin levels are higher while lower in skeletal muscle fibers than in healthy controls [106].

In addition, autophagy decreases during aging with progressive decline in muscle mass, strength, and quality [107]. Autophagic removal of damaged mitochondria appears to be one of the cellular mechanisms to prevent loss of muscle mass and quality by attenuating mitochondria-induced apoptosis in a healthy cell [108]. It would be crucial to have control over autophagy regulation in order to develop a cure for several diseases. For this reason, there has been increasing attention on the therapeutic modulation of autophagy, due to its cytoprotective nature [72,109,110,111]. If external retrograde control through the administration of humanin, autophagy, were possible, it would be an excellent therapeutic tool as it could be stimulated to increase normal cellular function and thus counteract senescence, as well as inhibited to treat cancer [72,112,113].

Sreekumar et al. [114] studied the expression of HN in human retinal pigment epithelial (hRPE) cells and its action on mitochondrial bioenergetics, oxidative stress-induced cell death, and senescence. Human RPE cells were treated with 150 μM tert-Butyl hydroperoxide (tBH) in the absence/presence of HN (0.5–10 μg/mL) for 24 h. Increased localization of exogenous humanin was detected in the mitochondrial and cytoplasmic compartments of hRPE cells. It was observed that humanin co-treatment inhibited the formation of ROS induced by tBH treatment and significantly restored mitochondrial bioenergetics in hRPE cells. Following treatment with humanin, an increase in mtDNA copies and upregulation of a key protein involved in mitochondrial biogenesis (mtTFA) were detected. HN was able to protect RPE cells from oxidative stress-induced cell death by STAT3 phosphorylation and inhibit caspase-3 activation. Thus, humanin treatment inhibited oxidant-induced senescence; in fact, it significantly reduced senescence-associated β-Gal-positive cells, ApoJ transcripts, and p16INK4a expression.

Other authors have demonstrated that HN increases long-lived protein degradation through stimulation of both basal and inducible autophagy [115]. The effect of humanin on intracellular protein degradation is not mediated by macro autophagy; instead, it was demonstrated that HN directly activates chaperone-mediated autophagy (CMA) and that activation of this pathway is required for the protective effect of HN against a variety of stressors including H_2_O_2_, PQ, rotenone, hypoxia, and starvation. CMA allows for the selective degradation of soluble proteins in lysosomes, contributing to the cellular quality control and maintenance of cellular energetic balance. Both exogenous and endogenous HN localize at the lysosomal membrane where they cooperate to enhance CMA efficiency. HN acts by stabilizing the binding of the chaperone HSP90 to the upcoming substrates at the cytosolic side of the lysosomal membrane. HN levels and the effectiveness of CMA both decrease with age. Interventions aimed at increasing HN levels could protect against oxidative stress by removing oxidative-damaged proteins through CMA activation.

Kim SJ et al. observed the effect of humanin on macro autophagy in four different study categories: cell lines, mice, *C. elegans*, and a human cohort [116]. It was observed that humanin induces autophagy in different cell types and in the skeletal muscle of aged mice. Autophagy is activated to maintain muscle integrity by eliminating protein aggregates and non-functioning mitochondria in skeletal muscle. This prevents the accumulation of damaged mitochondria that can induce oxidative damage and skeletal muscle apoptosis, and thus lead to muscle disease. It has already been reminded that in *C. elegans* transgenic worms with humanin, autophagy plays a critical role in the extension of lifespan. Moreover, humanin is co-expressed with autophagy-related genes in the human brain, further highlighting its effects on autophagy pathways. HN administration increases the expression of autophagy-related genes and reduces the accumulation of harmful misfolded proteins in the skeletal muscle of mice. Summing up, HN has a positive bioenergetic impact as evidenced through increased basal oxygen consumption rate, increased respiratory capacity, and increased ATP production in retinal pigment epithelial cells [117]. In the nude mole rat, a model of negligible senescence, it shows only a slight decrease in humanin levels during its exceptionally long lifespan of 30 years, supporting the idea that humanin is related to biological aging [30,40]. Part of the study was conducted on HEK293 cells to examine the induction of autophagy by various mitochondrial-derived peptides. The authors treated the cells with different doses of different MDPs to quantify autophagosomes and autolysosomes. They treated the cells with HNG, a potent analog of humanin, which has cytoprotective roles in various age-related diseases [116,118,119], proposed as a dietary restriction mimetic peptide by activating AMPK [120].

Humanin is known to activate chaperone-mediated autophagy by interacting with hsp90, and increasing substrate binding and translocation into lysosomes [115]. The results from several studies indicate that HNG increases autophagy via an upstream effector of PI3K class III complex formation such as ULK1, AMPK, and mTOR. HNG induces the autophagic process in cells by maintaining autophagic flux like humanin. It was important to understand whether HNG treatment increased autophagy proteins involved in the initiation complex (Beclin-1) and LC3 lipidation during elongation (ATG7, 5, and 3). It was evidenced that either 1 and 100 μM HNG augmented the expression of autophagy proteins involved in initiation and elongation. To study the role of humanin in the skeletal muscle of aged mice, HNG was administered to 18-month-old female C57Bl/6N mice for 14 months in order to understand its effects on lifespan and health [38], observing that it increases health span in mice by reducing inflammation and improving cognitive function [121], but it also increases mitochondrial biogenesis and AMPK pathways in skeletal muscle [122]. Furthermore, autophagy decreases with age together with multiple diseases while improving muscle function [107,123]; the hypothesis being that old HNG-treated mice are able to maintain their autophagy process better than the control group, thus improving mitochondrial and protein quality. The finding that humanin maintains the autophagic process and reduces abnormal protein accumulation in skeletal muscle during aging may be one of the mechanisms by which humanin-treated animals maintain better motor function while aging. The transgenic *C. elegans* HN worm increased the average lifespan from 24 days in the control to 27 days in the transgenic, demonstrating that autophagy plays an important role in humanin-induced lifespan extension. Furthermore, the results obtained from the study suggested that endogenous humanin could regulate autophagy pathways, as already demonstrated in the human brain, also in basal autophagy.

**Table 5 biology-12-00558-t005:** Role of Humanin in autophagy, N/A is not applicable.

Article	Study Design	Population	Outcome Measures
Miller, B. et al. (2022) [98]	Review	N/A	Aging; mitochondrial copy number; relative ratio of mtDNA to nuclear DNA, and autophagy
Li, P. et al. (2021) [107]	N/A	N/A	Autophagy in skeletal muscle
Sreekumar, P.G. et al. (2016) [114]	In vitro study	hRPE cells	Expression of humanin and its effect on oxidative stress-induced cell death; mitochondrial bioenergetics, and senescence.
Gong, Z. et al. (2018) [115]	In vitro study	N/A	Chaperone-mediated autophagy
Kim, S.J. et al. (2022) [116]	In vivo and in vitro study	HEK293 cells; Mouse; *C. elegans*; Human	Role of humanin in the activation and regulation of autophagy
Kim, S.J. et al. (2018) [124]	In vitro study	Primary senescent human fibroblasts	Number of mitochondria; levels of mitochondrial respiration; mtDNA methylation, and mitochondria-encoded peptides

Kim SJ et al. supposed that targeting metabolic pathways in senescent cells might be a novel strategy to eliminate senescent cells and modulate the senescence-associated secretory phenotype (SASP) [124]. They investigated the characteristics of mitochondria in doxorubicin-induced senescence using primary human fibroblasts rendered senescent by replicative exhaustion, hydrogen peroxide, or doxorubicin treatment.

Moreover, they examined the number of mitochondria and the levels of mitochondrial respiration, mtDNA methylation, and the mitochondria-encoded peptides of humanin and others. Senescent cells evidenced elevated levels of humanin, increased numbers of mitochondria, and higher levels of mitochondrial respiration, together with variable changes in mtDNA methylation. Exposure of senescent cells to high levels of humanin increased mitochondrial respiration and certain SASP factors. Additionally, fibroblast cells were treated with humanin for 10 days and no activity was found when examining SA-β-gal activity or SA-β-gal-positive cells. These results indicate that treatment with humanin not only does not cause senescence, but it might help senescent cells maintain a senescent status and the production of SASP factors.

#### 3.5.2. Cytoprotective Activity

Alsanousi et al. have revealed the correlation between the functional activity, tertiary structure, and partner recognition mode of humanin to better understand the molecular mechanisms of the cytoprotective activity of HN [125]. Already, some studies, Table 6, had highlighted the relationship between the cytoprotective activity that the optical isomerization of Ser14 residue from the l to d form without, however, explaining the details of the molecular mechanism. The authors identified an Aβ-binding site on humanin D-Ser14 by NMR, detecting the structure-function relationships of HN and explaining the mechanisms of the conformational change favoring the interaction caused by d-isomerization of residue Ser14. Purified peptides (>95% purity) from wild-type HN, HNS14G, and HN D-Ser14 were used for the study. To identify which among the three variants had higher affinity, they studied the Aβ-binding sites of HN and HN DSer14 showing which of them had higher inhibitory activity toward fibril formation given by Abeta aggregation/fibrillation. HN S14G and HN DSer14 were shown to have greater cytoprotective activity than HN and zinc ions which could demonstrate the aggregation pathway of HN variants and explain how HN suppresses zinc-mediated Aβ aggregation or fibrillation to become a future site of exogenous action.

HN-dependent citoprotection is mediated in part by interacting with and antagonizing pro-apoptotic Bax-related peptides and IGFBP-3 (IGF-binding protein 3). An in silico study identified six additional peptides in the same region of mtDNA as humanin, which were named small human-like peptides (SHLPs) [14]. In particular, the authors investigated the effects of SHLP2 and SHLP3 on apoptosis and cell metabolism. Like HN, both SHLP2 and SHLP3 improved cell survival in response to toxic insults, reduced the generation of ROS, prevented apoptosis, and improved mitochondrial metabolism in vitro, while SHLP6 had the opposite effect. Systemic hyperinsulinemic–euglycemic clamp investigations indicated that intracerebrally infused SHLP2 increased glucose uptake and suppressed hepatic glucose production, suggesting that it functions as an insulin sensitizer both peripherally and centrally. In analogy with HN, also the concentrations of circulating SHLP2 were found to decrease with age. All these evidences, together with the neuroprotective effect and the anti-oxidative stress action of SHLP2, imply that SHLP2 has a role in the regulation of aging and age-related conditions, such as Alzheimer’s disease, diabetes, and atherosclerosis [14].

As it has already been extensively discussed, HN inhibits the binding to the mitochondrial membrane, and the oligomerization of Bax and Bid proteins, which would reduce their permeability. On the other hand, both Bax and Bid are potential mediators of the pro-apoptotic effects of GC in chondrocytes [126], which is why the authors hypothesized that HN, and its potent analog HNG, could prevent the long-term effects of glucocorticoids such as impaired bone growth. During the study, the researchers showed that HN does not interfere with the anti-inflammatory effects of these molecules as a new regulator of Hedgehog signaling. This phenomenon was observed in a growth plate showing longitudinal bone growth of chondrocytes and their proliferation in response to various hormones [127], and substances such as dexamethasone (Dexa), a glucocorticoid. Normally, glucocorticoids alter the proliferation/differentiation of chondrocytes and disrupt the normal growth of cartilage lengthwise [128].

**Table 6 biology-12-00558-t006:** Cytoprotective effect of humanin, N/A is not applicable.

Article	Study Design	Population	Outcome Measures
Alsanousi, N. et al. (2016) [125]	In vitro study	N/A	Inhibitory effect against amyloid-β fibrillation of humanin
Zaman, F. et al. (2019) [126]	In vitro study	N/A	Humanin regulator of Hedgehog signaling and prevents glucocorticoid-induced bone growth impairment
Qin, Q. et al. (2018) [129]	In vivo study	Mouse	Effect of exogenous humanin to prevent and reverse cardiac fibrosis and apoptosis in the aging heart
Liu, C. et al. (2019) [106]	In vivo study	Human	Expression levels of humanin and MOTS-C in skeletal muscle and serum levels in CKD

A key role in growth and as a regulator of proliferation/differentiation is played by Indian Hedgehog factor (Ihh) secreted by chondrocytes [14,15]. HN has already been shown in mouse models to protect against GC-induced bone growth impairment by Bax ablation [126], has shown anti-inflammatory effects, and has presented itself as a regulator of Hh signaling that prevents GC-induced bone growth impairment without interfering with the desired actions of GCs. This in vitro finding nominated humanin as a possible therapeutic substance that counteracts the action of glucocorticoids, especially in growing subjects [126].

Several authors have tried to understand what role humanin might play in the various tissue districts. Qin et al. tested the effect of exogenous humanin to prevent and reverse cardiac fibrosis and apoptosis in the aging heart of middle-aged mice. Female C57BL/6N mice at 18 months of age received 14-mo intraperitoneal injections of vehicle (old group; *n* = 6) or HN analog (HNG; 4 mg/kg 2 times/wk, old + HNG group, *n* = 8) and were euthanized at 32 months of age. C57BL/6N female mice (young group, *n* = 5) at 5 months of age were used as young controls. In mice that received HNG treatment, an increase in the ratio of cardiomyocytes to fibroblasts in aging hearts was observed. HNG treatment significantly reduced the enhanced interstitial collagen deposition in aged hearts, together with cardiac fibroblast proliferation, and attenuated fibroblast growth factor-2, matrix metalloproteinase-2 expression, and transforming growth factor-β1 in aging mice. Additionally, myocardial apoptosis was inhibited in HNG-treated aged mice. Data from this study indicate that the cardioprotective effect of HNG is associated with activation of Akt/GSK-3β signaling as well as an alteration in pro-apoptotic factors (Bax, Bid, and Bim). Administration of HNG activated the Akt pathway. Activated Akt inactivates pro-apoptotic proteins, including Bax, Bad, Bim, Bid, and caspase-9. Moreover, activated Akt directly phosphorylates GSK-3β at Ser9, which negatively regulates its kinase activity, inhibits the opening of mitochondrial permeability transition pores, and inhibits myocardial apoptosis and fibrosis in heart failure. Moreover, HN mediates its anti-apoptotic effect through intracellular pathways. In fact, HN binds to pro-apoptotic proteins (Bax, Bid, and Bim), inactivating their function [129].

Among the pathologies characterizing senescence, advanced chronic kidney disease (CKD) is characterized by a premature aging phenotype of multifactorial origin. Mitochondrial dysfunction has been proposed, for example, as one of the main factors contributing to poor muscle function. Liu C et al. examined the expression levels of humanin in skeletal muscle and serum levels in CKD at stage 5 patients compared to patients with normal renal function. High circulating levels of humanin were observed while local muscle levels were reduced. Humanin in serum correlated positively to circulating TNF levels, a marker of systemic inflammation in CKD. Reduced humanin and MDP levels, in general, in skeletal muscle were associated with lower mitochondrial density and evidence of oxidative stress. The results of this study indicate a differential regulation of MDPs in CKD and suggest an alternative site of production for circulating humanin than skeletal muscle. Reduced skeletal muscle MDP levels were inversely related to systemic inflammation and oxidative stress [106].

## 4. Conclusions

Humanin, a peptide encoded by mtDNA, plays a cytoprotective and senolytic role, helping to preserve mitochondrial function and cell viability under conditions of senescent cell-associated diseases and stress, and is potentially a drug candidate for a number of age-related diseases, proving to be a fruitful avenue for future studies and possible developments.

## Figures and Tables

**Figure 2 biology-12-00558-f002:**
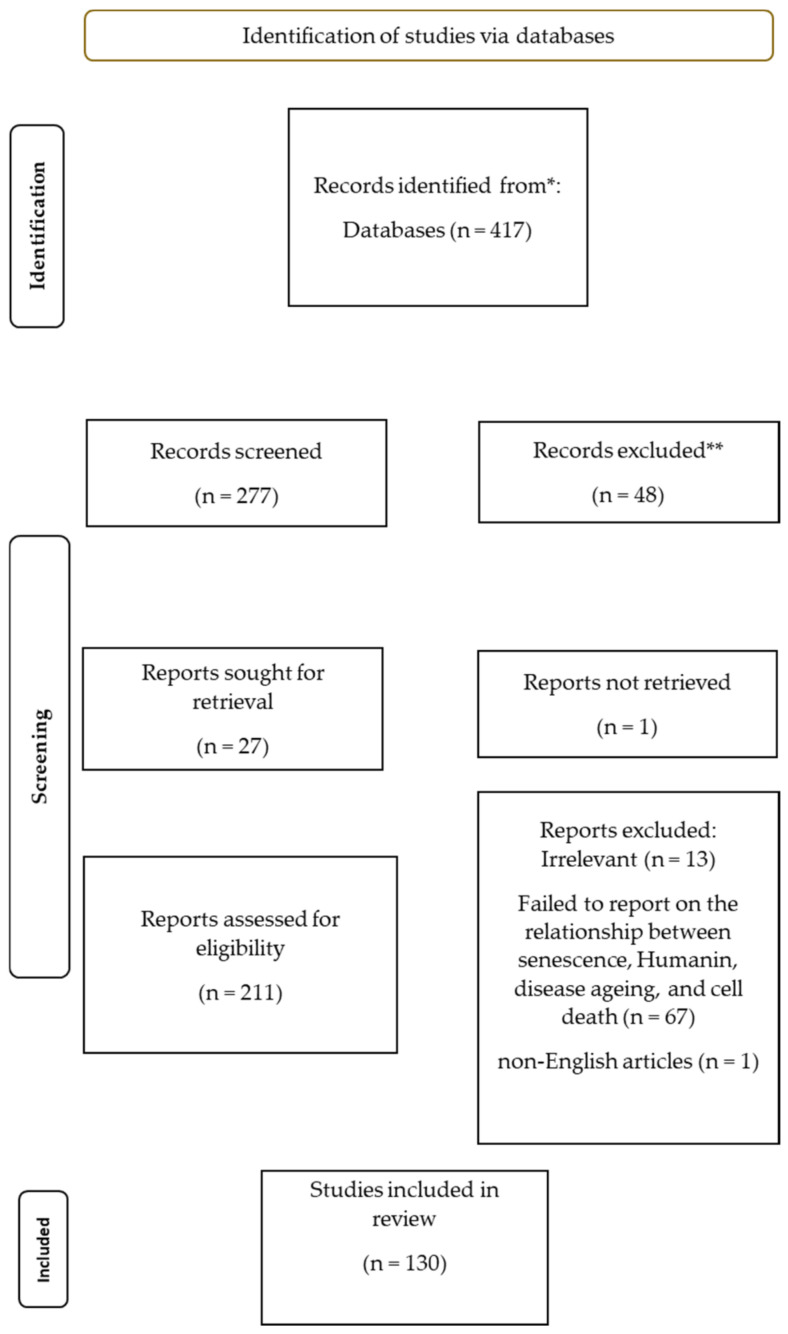
PRISMA flow diagram; *n* is the number of articles selected, * total number, ** other topics.

**Table 1 biology-12-00558-t001:** Neuroprotective effect of Humanin, N/A is not applicable.

Article	Study Design	Population	Outcome Measures
Zárate, S.C. et al. (2019) [50]	In vivo study	Rat	Neuroprotective effect of humanin and relationship with ovarian hormones
Yen, K. et al. (2018) [38]	In vitro and in vivo study	SH-SY5Y cells, Mouse	Neuroprotective effect of humanin
Yen, K. et al. (2020) [40]	In vivo study	*C. elegans*, Mouse, Human	Circulating levels of humanin and their relation to diseases of aging and lifespan
Gong, Z. et al. (2022) [49]	Review	Mouse, Porcine	Protective effect of humanin in myocardial ischemia-reperfusion.
Kim, S.J. et al. (2021) [37]	Review	N/A	Humanin in age-related disease
Gong, Z. et al. (2014) [51]	Review	N/A	Role of humanin in age-related disease
Caso, V.M. et al. (2021) [42]	In vitro study	N/A	Neuroprotective effects of humanin and its homologs (HNG) from Aβ
Zacharias, D.G. et al. (2012) [43]	In vivo study	Mouse	Humanin reduced plaque accumulation in Alzheimer’s disease and has a cytoprotective action in stroke
Park, T. et al. (2013) [55]	In vivo study	Middle-aged APPswe/PS1dE9 mice	Treatment with HNG significantly improves spatial learning and memory deficits, reduces Aβ plaque accumulation and insoluble Aβ concentration; decreases neuro-inflammatory responses

**Table 3 biology-12-00558-t003:** Role of Humanin in inflammation, N/A is not applicable.

Article	Study Design	Population	Outcome Measures
Esterhuizen, K. et al. (2017) [80]	In vitro study	N/A	Role of HN against oxidative stress, apoptosis, and inflammatory response
Bachar, A.R. et al. (2010) [82]	In vitro study	N/A	HN attenuates inflammation and macrophage infiltration, reduces in vitro production of IL-6, IL-1β, and TNF-α
Nashine, S. et al. (2022) [94]	In vivo study	AMD patients AMD RPE transmitochondrial cybrid cells	Treatment with HNG can reduce plasma levels of inflammation-associated marker protein. In AMD RPE cybrid cells, treatment with HNG reduced CD62E/E-Selectin, CD62P/P-Selectin, ICAM-1, TNF-α, MIP-1α, IFN–γ, IL-1β, IL-13. and IL-17A
Conte, M. et al. (2021) [89]	In vivo study	Patients with T2D and AD	Plasma levels of fibroblast growth factor 21 (FGF21), growth differentiation factor 15 (GDF15), humanin, and mitochondrial stress-related mitokines
Conte, M. et al. (2019) [30]	In vivo study	Human	Aging; longevity; plasma levels of different mitokines
Merry, T.L. et al. (2020) [95]	In vivo study	Human	Plasma humanin levels in long-lived subjects

**Table 4 biology-12-00558-t004:** Role of Humanin in diabetes and metabolism, N/A is not applicable.

Article	Study Design	Population	Outcome Measures
Boutari, C. et al. (2022) [46]	Review	N/A	Role of humanin in age-related diseases
Mehta, H.H. et al. (2019) [96]	In vivo study	DIO Mouse	Humanin effect on metabolism
Merry, T.L. et al. (2020) [95]	Review	N/A	Role of humanin in metabolism; relationship with (GH)/IGF-1
Muzumdar, R.H. et al. (2009) [29]	In vivo study	Rat	Central effects of HN on insulin action

## Data Availability

Not applicable.

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
