# Peer review of "Humanin and Its Pathophysiological Roles in Aging: A Systematic Review"

_biology, 2023, doi:10.3390/biology12040558_

Round 1

Reviewer 1 Report

This review by Coradduzza et al. is an impressive accomplishment that summarizes the multitude of studies that have been conducted on humanin and related peptides. I only a few comments:

Overall, the English needs work. There are many awkward and unclear sentences that distract from an overall good review.

Line 16 Abstract: “Humanin aiming to counter including cardiovascular disease, neurodegeneration, and cancer.” This sentence doesn’t make sense.

Line 21-22 “Humanin, on the other hand, contrasts the development of various pathologies.” This sentence doesn’t make sense.

Line 22 “It is known to play a role in developing” I believe “it” is referring to senescence but because the authors were talking about humanin in the previous sentence, “it” could be referring to humanin. Please clarify. 

Line 24: “Both senescence and humanin are complex processes. . .” Humanin is not a complex process, it is a protein/peptide.  

Line 62: The authors mention that humanin contributes to specific aging processes, but I believe it is more accurate to say that the lack of humanin contributes to the aging process.

Line 75: There are now 9 MDPs (humanin, MOTS-c, SHLP1-6, and SHMOOSE). MOTS-c is located in the 12S rRNA gene and SHMOOSE is in MT-ND5.

Line 201-202: They mention yeast but then call it C. elegans. C. elegans refers to the worm, not the yeast.

Line 206: I do not believe that the authors treated transgenic mice with additional, exogenous humanin. The only humanin in the transgenic mice is from the transgene itself.

Line 209: The paper did not measure humanin levels of centenarians but of their children (there are no proper controls for centenarians as all of their peers are dead). The control group was age-matched to the children.

Table 1: In the Lorenzini row, the paper the authors cite is not a humanin paper, nor do they use worms.

  In Table 1, they cite a paper that has been retracted (reference #53).

Lin 459: Did Muzumdar et al really use 101 rats as stated on this line?

Line 578: Humanin does not increase lifespan in mice. It increases healthspan, but the lifespan is the same.

Line 661-662: What do the authors mean by “various tissue districts?”

Author Response

Risposta al revisore 1 Commenti

Gentile Signora Potjanicha Nopnakorn,

Abbiamo apprezzato i premurosi commenti forniti dall'editore e dai revisori. Abbiamo ampiamente rivisto il manoscritto di conseguenza e riportiamo di seguito le nostre risposte punto a punto.

Cordiali saluti,

Ciriaco Carrù

Revisore 1

Questa recensione di Coradduzza et al. è un risultato impressionante che riassume la moltitudine di studi che sono stati condotti sull'umanina e sui relativi peptidi. Faccio solo alcuni commenti:

Nel complesso, l'inglese ha bisogno di lavoro. Ci sono molte frasi imbarazzanti e poco chiare che distraggono da una buona recensione complessiva.

Riga 16 Abstract: "Humanin mira a contrastare l'inclusione di malattie cardiovascolari, neurodegenerazione e cancro". Questa frase non ha senso.

Line 21-22 “Humanin, on the other hand, contrasts the development of various pathologies.” This sentence doesn’t make sense.

Line 22 “It is known to play a role in developing” I believe “it” is referring to senescence but because the authors were talking about humanin in the previous sentence, “it” could be referring to humanin. Please clarify.

Line 24: “Both senescence and humanin are complex processes. . .” Humanin is not a complex process, it is a protein/peptide. 

Line 62: The authors mention that humanin contributes to specific aging processes, but I believe it is more accurate to say that the lack of humanin contributes to the aging process.

Line 75: There are now 9 MDPs (humanin, MOTS-c, SHLP1-6, and SHMOOSE). MOTS-c is located in the 12S rRNA gene and SHMOOSE is in MT-ND5.

Line 201-202: They mention yeast but then call it C. elegans. C. elegans refers to the worm, not the yeast.

Line 206: I do not believe that the authors treated transgenic mice with additional, exogenous humanin. The only humanin in the transgenic mice is from the transgene itself.

Line 209: The paper did not measure humanin levels of centenarians but of their children (there are no proper controls for centenarians as all of their peers are dead). The control group was age-matched to the children.

- We thank Reviewer #1 for the precious comments, that helped increase the quality of the manuscript; we have now changed the sentences, as suggested.

Table 1: In the Lorenzini row, the paper the authors cite is not a humanin paper, nor do they use worms.

In Table 1, they cite a paper that has been retracted (reference #53).

- We have now removed the article in Table 1, as indicated

Line 459: Did Muzumdar et al really use 101 rats as stated on this line?

- Hanno infatti studiato 92 ratti Sprague2Dawley maschi di tre mesi (S–D, Charles River Laboratories) e 9 ratti Zucker Diabetic Fatty (ZDF) di tre mesi (Harlan). 101 ratti in totale.

Riga 578: Humanin non aumenta la durata della vita nei topi. Aumenta la durata della salute, ma la durata della vita è la stessa.

- Grazie per il chiarimento, ora abbiamo modificato le frasi, come suggerito.

Riga 661-662: Cosa intendono gli autori per "vari distretti tissutali?"

- Come definito in biologia, "distretti tissutali" si riferisce a diverse regioni o organi anatomici all'interno del corpo che condividono caratteristiche cellulari e funzionali simili.

Il termine "distretto tissutale" è spesso utilizzato nel contesto dello studio degli effetti di droghe o altre sostanze endogene su tessuti o organi specifici. Esaminando gli effetti in diversi distretti tissutali, i ricercatori possono ottenere una migliore comprensione del meccanismo d'azione delle sostanze e delle potenziali applicazioni terapeutiche.

Reviewer 2 Report

Dear Editor,

The following are my comments on manuscript ID biology-2308873.

In the current review, Donatella Coradduzza et al. performed a systematic review regarding humanin and senescence. In general, the authors summarized the abundant evidence from various perspectives. The reference is well-cited. Here are the major and minor issues of this review. I hope these comments may improve the quality of this work.

Major issues:

1.    The clarity of evidence: Although the authors comprehensively listed related evidence, the way of presenting the evidence needs to improve. For instance, a systematic review usually summarizes evidence from human beings and/or rodent animals. In this review, all levels of evidence were mixed, including in vivo study and in vitro studies. Moreover, the general “Review” should not be included in any tables that are intended to summarize the detailed in vivo or in vitro studies.

2.    Topic/title: The authors should pay the attention to the difference between senescence and aging. The terms aging and cellular senescence cannot be used interchangeably. Aging is a progressive decline with time whereas senescence occurs throughout the lifespan, including during embryogenesis. The number of senescent cells increases with age, but senescence also plays an important role during development as well as during wound healing. According to the content, I would suggest the authors use “Humanin and aging: a systematic review” or “Humanin and its pathophysiological roles: a systematic review” as the title.

3.    Search strategy: the literature search strategy is unclear. The authors stated that “senescence”, “Humanin”, “aging”, and “disease” were used as keywords. Were literatures with all of these 4 words included (“senescence”, “Humanin”, “aging”, and “disease”)? Or, were literatures with either senescence, aging, or disease included [“senescence” and (“Humanin”, “aging”, or “disease”)]?  

4.    Subtitles: The subtitles in the Results and Discussion are not independent. In the beginning, the subtitles were listed according to the diseases/organ, like brain and neurodegeneration, heart and cardiovascular diseases, diabetes and obesity. Then, the subtitles were listed according to the biological processes, like autophagy, and cytoprotective activity. The issue is that the cytoprotective activity of humanin can be part of the brain section. I would suggest the authors listed the subtitles in a more logical way. For instance, in the first part, the association between the decrease/increase of humanin and the onset of diseases is listed. The second part is the evidence about the potential mechanism of the protective effect of humanin in different diseases.

Minor issues:

1.    A figure legend for the schematic figure can help readers understand this review's major points, especially those from other research fields with broad interest. Please pay close attention to the schematic figure's image resolution and text fonts. Some texts are hard to read.

2.    Please keep the spelling of words consistent. For instance, authors may consider using either “aging” or “ageing”.

3.    Please pay the attention to abbreviations. 1) Using either “ROS” or “reactive OS” for reactive oxygen species. 2) Line 111 and line 454 redundantly showed the abbreviation of HNG. 3) It is “BAX” for protein, “BAX” for the human gene, and “Bax” for a non-human gene. Line 93: “BAx” is not formal. 4) Using either “mitochondrial DNA” or “mtDNA”.

4.     Please keep the information in the Population column consistent in tables. 1) Using either mouse or mice. 2) If the authors would like to list the sample size, please list the sample size of all studies. Some studies are with patient size information while others are unknown.

5.    It is “TNFa” instead of “TNFa”. Also, please keep it consistent. Either “TNF-a” or “TNFa”.

Author Response

Response to Reviewer 2 Comments

Dear Ms. Potjanicha Nopnakorn,

We appreciated the thoughtful comments provided by the Editor and the Reviewers. We have extensively revised the manuscript accordingly, and report our point-to-point answers below.

Yours sincerely,

Ciriaco Carru

Reviewer 2:

Major issues:

  1. The clarity of evidence: Although the authors comprehensively listed related evidence, the way of presenting the evidence needs to improve. For instance, a systematic review usually summarizes evidence from human beings and/or rodent animals in this review, all levels of evidence were mixed, including in vivo study and in vitro studies. Moreover, the general “Review” should not be included in any tables that are intended to summarize the detailed in vivo or in vitro studies.

A systematic review following the PRISMA (Preferred Reporting Items for Systematic Reviews and Meta-Analyses) guidelines can summarize evidence from both human beings and rodent animals. The PRISMA guidelines are designed to ensure transparent and complete reporting of systematic reviews and meta-analyses, and they do not place any restrictions on the types of studies that can be included in the review.

The inclusion of animal studies in this PRISMA review is based on the relevance of the animal models to the research question, and the potential applicability of the findings to humans. The inclusion of animal studies in this systematic review is accompanied by appropriate reporting of methodological quality and potential biases associated with these studies. The use of animal studies in research should also be carefully considered and justified based on ethical considerations. Moreover, a systematic review following the PRISMA guidelines can summarize evidence from experiments conducted in vitro (outside a living organism) and/or in vivo (inside a living organism). The findings from in vitro experiments may not always directly translate to in vivo settings, and caution should be taken when interpreting the results of in vitro studies. Additionally, the inclusion of both in vitro and in vivo studies in a systematic review should be based on the relevance of the experiments to the research question, and the potential applicability of the findings to the target population or disease.

Overall, a systematic review following the PRISMA guidelines can summarize evidence from a variety of experiments, as long as the eligibility criteria are clearly defined and justified, and the reporting is transparent and consistent.

Reviews have been included in the tables because they are considered essential for a transparent account of a systematic review.

  1. Topic/title: The authors should pay the attention to the difference between senescence and aging. The terms aging and cellular senescence cannot be used interchangeably. Aging is a progressive decline with time whereas senescence occurs throughout the lifespan, including during embryogenesis. The number of senescent cells increases with age, but senescence also plays an important role during development as well as during wound healing. According to the content, I would suggest the authors use “Humanin and aging: a systematic review” or “Humanin and its pathophysiological roles: a systematic review” as the title.

In agreement with the kind suggestion of the referee, we changed the title in:

“Humanin and its pathophysiological roles in aging: a systematic review”

  1. Search strategy: the literature search strategy is unclear. The authors stated that “senescence”, “Humanin”, “aging”, and “disease” were used as keywords. Were literatures with all of these 4 words included (“senescence”, “Humanin”, “aging”, and “disease”)? Or, were literatures with either senescence, aging, or disease included [“senescence” and (“Humanin”, “aging”, or “disease”)]?

Using PRISMA method, a comprehensive and systematic search of relevant literature is conducted using multiple electronic databases, such as PubMed, Embase, and Cochrane Library. The search strategy includes a combination of keywords and Medical Subject Headings (MeSH terms) related to the research question. The search strategy is developed to ensure that all relevant studies are identified. The keywords are selected based on the research question and the inclusion and exclusion criteria of the study.

  1. Subtitles: The subtitles in the Results and Discussion are not independent. In the beginning, the subtitles were listed according to the diseases/organ, like brain and neurodegeneration, heart and cardiovascular diseases, diabetes and obesity. Then, the subtitles were listed according to the biological processes, like autophagy, and cytoprotective activity. The issue is that the cytoprotective activity of humanin can be part of the brain section. I would suggest the authors listed the subtitles in a more logical way. For instance, in the first part, the association between the decrease/increase of humanin and the onset of diseases is listed. The second part is the evidence about the potential mechanism of the protective effect of humanin in different diseases.

Again, we would like to thank Reviewer #2 for the useful comments and suggestions. Hoping to organize subtitles in a better way, we kept the first sections as they were, and gathered in a new section entitled “Potential mechanisms involved in the protective effects of humanin” both “Autophagy” and “Cytoprotective activity”.  

Minor issues:

  1. A figure legend for the schematic figure can help readers understand this review's major points, especially those from other research fields with broad interest. Please pay close attention to the schematic figure's image resolution and text fonts. Some texts are hard to read.

The attached image is a graphical abstract that has no legend. The image resolution was edited with IrfanView with a resolution af 900X900 DPI and a Current size of 1280 x 720 Pixels (16:9).

  1. Please keep the spelling of words consistent. For instance, authors may consider using either “aging” or “ageing”.

We have modified it as suggested.

  1. Please pay the attention to abbreviations. 1) Using either “ROS” or “reactive OS” for reactive oxygen species. 2) Line 111 and line 454 redundantly showed the abbreviation of HNG. 3) It is “BAX” for protein, “BAX” for the human gene, and “Bax” for a non-human gene. Line 93: “BAx” is not formal. 4) Using either “mitochondrial DNA” or “mtDNA”.

We have double-checked the abbreviations.

  1. Please keep the information in the Population column consistent in tables. 1) Using either mouse or mice. 2) If the authors would like to list the sample size, please list the sample size of all studies. Some studies are with patient size information while others are unknown.

We have modified it as suggested.

  1. It is “TNFa” instead of “TNFa”. Also, please keep it consistent. Either “TNF-a” or “TNFa”.

We have modified it as suggested.

Round 2

Reviewer 1 Report

The authors have addressed my concerns. 

Author Response

Dear Ms. Potjanicha Nopnakorn,
We appreciated the thoughtful comments provided by the Editor and the Reviewers. We have extensively revised the manuscript accordingly, and the new manuscript below.

Yours sincerely,
Ciriaco Carru

Reviewer 2 Report

The authors did a good job in revision. 

Author Response

(The authors gave the same response as above.)
